# Trafficking and Function of the Voltage-Gated Sodium Channel β2 Subunit

**DOI:** 10.3390/biom9100604

**Published:** 2019-10-13

**Authors:** Eric Cortada, Ramon Brugada, Marcel Verges

**Affiliations:** 1Cardiovascular Genetics Group, Girona Biomedical Research Institute (IDIBGI), C/ Doctor Castany, s/n—Edifici IDIBGI, 17190 Girona, Spain; ecortada@gencardio.com (E.C.); rbrugada@idibgi.org (R.B.); 2Biomedical Research Networking Center on Cardiovascular Diseases (CIBERCV), 28029 Madrid, Spain; 3Medical Sciences Department, University of Girona Medical School, 17003 Girona, Spain; 4Cardiology Department, Hospital Josep Trueta, 17007 Girona, Spain

**Keywords:** cardiac arrhythmias, protein trafficking, voltage-gated sodium channel, Na_V_1.5, *SCN2B*

## Abstract

The voltage-gated sodium channel is vital for cardiomyocyte function, and consists of a protein complex containing a pore-forming α subunit and two associated β subunits. A fundamental, yet unsolved, question is to define the precise function of β subunits. While their location in vivo remains unclear, large evidence shows that they regulate localization of α and the biophysical properties of the channel. The current data support that one of these subunits, β2, promotes cell surface expression of α. The main α isoform in an adult heart is Na_V_1.5, and mutations in *SCN5A*, the gene encoding Na_V_1.5, often lead to hereditary arrhythmias and sudden death. The association of β2 with cardiac arrhythmias has also been described, which could be due to alterations in trafficking, anchoring, and localization of Na_V_1.5 at the cardiomyocyte surface. Here, we will discuss research dealing with mechanisms that regulate β2 trafficking, and how β2 could be pivotal for the correct localization of Na_V_1.5, which influences cellular excitability and electrical coupling of the heart. Moreover, β2 may have yet to be discovered roles on cell adhesion and signaling, implying that diverse defects leading to human disease may arise due to β2 mutations.

## 1. Introduction

We often wonder why people get sick. Why do some people die early in life while others reach old age without major health problems? It is obvious that both genetic and environmental factors are playing a role. An organ that must certainly remain in perfect condition throughout life is the heart; its abnormal functioning is a sign of diseases that can lead to premature death. An unsolved matter is to determine the molecular alterations that can lead to heart disease and how these may arise.

### 1.1. Arrhythmias and Sudden Cardiac Death (SCD)

Cardiomyopathies are a type of disease affecting the function of the heart muscle and leading to heart failure. Suffering from cardiomyopathy also implies being at risk of arrhythmia and sudden cardiac death (SCD). In the US alone, SCD affects annually up to 400,000 people, with coronary heart disease being present in 80% of the cases [1]. Importantly, it is estimated that around 6 million people die each year worldwide of SCD due to ventricular arrhythmia, which is often associated with cardiomyopathy (the survival rate of a cardiac arrest is <1%). This represents 1/3rd of deaths from cardiovascular disease of any kind, being these in fact the main cause of death in the world [2]. Therefore, understanding how arrhythmias develop, and the consequences that they entail, has interest in the fields of biology, medicine and socioeconomics.

SCD can occur in people under 50 and be associated with pathologies defined as unexplained or arrhythmogenic. It is often the first and single clinical manifestation of an inherited heart disease that has remained undiagnosed by conventional clinical practice [3]. A well-known arrhythmia is Brugada syndrome (BrS), a disorder characterized by an abnormal electrocardiogram (ECG) that causes ventricular fibrillation. Although considered a rare genetic disease, it is actually inherited by an autosomal dominant trait [4] and linked to a high mortality rate [5]. In BrS, the ventricular muscle quivers, instead of contracting in a coordinated manner. Such tachycardia prevents the blood from flowing efficiently throughout the body. Consequently, the individual faints and can die within a few minutes [6].

Since genetics and molecular biology emerged in the field of clinical cardiology, multiple genes and mutated variants responsible for arrhythmias that lead to SCD were identified. A fundamental hypothesis is that many cases of SCD are the clinical manifestation of rare hereditary diseases caused by mutations in transmembrane ion channels implicated in generating the cardiac action potential (AP). These channelopathies are due to alterations in subunits of the channel protein complex, as well as in associated or regulatory proteins. Therefore, they can be congenital and due to one or more mutations in the genes involved. Along with BrS, other channelopathies that cause SCD include long QT syndrome (LQTS) and catecholaminergic polymorphic ventricular tachycardia [7].

### 1.2. The Cardiac Voltage-Gated Sodium (Na_V_) Channel

The voltage-gated sodium (Na_V_) channel often shows alterations leading to cardiac channelopathies [8]. For instance, ~20 % of BrS cases are caused by mutations in *SCN5A*, a gene mapping to the chromosomal region 3p21 [9] and encoding Na_V_1.5, the pore-forming, α subunit, of the main cardiac Na_V_ channel [4]. In the heart, the Na_V_ channel is responsible for generating the rising phase of the AP, thus playing a central role in myocardial excitability. The abnormal ECG observed in BrS is due to loss-of-function of the Na_V_ channel. It is characterized by an ST segment elevation of the V1-V3 precordial leads and occurs in the absence of structural heart disease [6] (Figure 1).

Initiation of the AP, along with the degree of intercellular communication via gap junctions, to allow propagation, determines the conduction velocity of the electrical impulse. Thus, mutations in *SCN5A* have also been associated with LQTS, atrial fibrillation and even cardiomyopathies [10]. In fact, the Na_V_ channel also plays an important role in electrical impulse propagation in the heart [11].

Na_V_1.5 is located in the sarcolemma, i.e., the cardiomyocyte plasma membrane. Its localization and function are regulated by auxiliary β subunits and other associated proteins [12,13]. Alterations in the channel biophysical properties —including changes in activation or inactivation kinetics—can give rise to the channel’s gain- or loss-of-function [8]. In this context, potentially fatal arrhythmias may be due to defects in regulation of transcription, affecting *SCN5A* expression, or to post-translational modifications, causing, for example, an incomplete processing of Na_V_1.5 that may turn into its retention in the exocytic pathway. A precise subcellular location of Na_V_1.5 is essential for the channel role in AP generation (Figure 2). Thus, defects in Na_V_1.5 targeting to the sarcolemma—or an incorrect or inadequate anchoring at the membrane—can affect considerably Na_V_ channel functioning. In fact, alterations in these processes are associated with BrS [14].

To understand the role of the Na_V_ channel in cardiac function, and the consequent alterations that may lead to disease, it is essential to study traffic and localization of Na_V_1.5, and importantly, the contribution of its associated and regulatory proteins. In fact, considerable effort has been made to investigate the pathways that determine the correct location of Na_V_1.5 to subregions of the sarcolemma. It is known that several proteins interact with Na_V_1.5, including cytoskeleton components, and various structural domains involved in these interactions have been identified [13,15,16]. In addition, it has been reported the assembly of macromolecular complexes between Na_V_1.5 and the inward rectifier potassium channel Kir2.1 (which contributes to the final repolarization phase of the AP, and controls the resting membrane potential of ventricular cardiomyocytes), which localize together in microdomains of the sarcolemma, thereby controlling excitability [17]. Interestingly, an unconventional anterograde route between Na_V_1.5 and Kir2.1/2.2 has been described, which would allow reciprocal regulation of their localization. Understanding these interactions should help determining defects in localization of these channels at the sarcolemma. This is important to subsequently assess the clinical features and severity of the disease. For instance, depending on the BrS-associated mutation in *SCN5A*, a different pathological retention of Na_V_1.5 may be observed, either in the endoplasmic reticulum (ER) or in the Golgi apparatus. In this regard, ER-, but not Golgi-retained Na_V_1.5 mutants, can be partially restored by Kir2.1/2.2, thereby ameliorating the I_Na_ reduction due to the *SCN5A* mutation. Thus, Golgi-retained Na_V_1.5 mutants give a comparatively higher decrease in I_Na_, and therefore of cardiac excitability, than the ER-retained mutants [14].

Na_V_1.5 is located at intercalated discs (ID), that is, the region between cardiomyocytes containing cell adhesion complexes, and also in the lateral membrane [13], where costameres and transverse tubules (T tubules) are found; costameres are protein complexes connecting the cardiomyocyte sarcomeres with the extracellular matrix, while T tubules are deep invaginations of the sarcolemma in the Z-disk region (or Z-line, which borders the sarcomeres), allowing electrical coupling with Ca^2+^ release from the sarcoplasmic reticulum [18]. The distribution of Na_V_1.5 in ID and T tubules depends in part on ankyrin-G (AnkG) [19]. In fact, multiple interactions implicating Na_V_1.5 have been described, which establish protein complexes defining different pools of Na_V_1.5 in cardiomyocytes (Figure 3). Thus, plakophilin 2 (PKP2) and SAP97 (synapse-associated protein 97; a member of the family of membrane-associated guanylate kinases) would control Na_V_1.5 localization at the ID [20]. It has been proposed that PKP2 is part of a complex with connexin 43 (Cx43) and AnkG, possibly independent of the interaction with SAP97 [21]. Na_V_1.5 is also located in the lateral membrane, whose targeting is regulated by the syntrophin/dystrophin complex connected to the actin cytoskeleton [20]. AnkG is also associated with Na_V_1.5 in T tubules [22]. Yet, AnkG is clearly required for Na_V_1.5 targeting to the ID [23]. Finally, it was shown by super-resolution microscopy that a complex formed by Cx43, PKP2 and N-cadherin is responsible for the delivery of microtubule cargo to the ID, including Na_V_1.5; the components and location of this molecular complex define the so-called connexome, which regulates electrical coupling, cell adhesion, and cell excitability [24].

Understanding the distribution of Na_V_1.5 in sarcolemma subdomains has indeed become a complex task. Thus, recent data suggest the presence of a sub-pool of Na_V_1.5 at the lateral membrane that is independent of syntrophin [25]. Interestingly, even the associated β1 subunit (see below) appears differentially localized at sarcolemma subdomains. In this regard, Tyr-phosphorylated β1 (pYβ1) is associated with Na_V_1.5 at the ID, along with other partners, such as AnkG, Cx43 and N-cadherin, whereas non-phosphorylated β1 associates with Na_V_ channels in T tubules [26].

In summary, Na_V_1.5 is probably grouped into well-defined functional nanodomains in each subregion [27]. In fact, Na_V_1.5 has been found associated with lipid rafts, which are membrane domains rich in cholesterol and glycosphingolipids where ionic channel regulatory proteins concentrate [18]. Understanding how all these molecules interact to regulate the subcellular localization of Na_V_ channels is clearly a challenge in this field of research.

### 1.3. The Na_V_ Channel β Subunits

Analysis of Na_V_1.5 trafficking can be envisaged from at least three standpoints. First, we must understand how Na_V_1.5 is targeted to cell surface domains; secondly, how Na_V_1.5 is retained in certain subregions or domains of the sarcolemma, thereby defining its surface distribution; and third, how Na_V_1.5 endocytosis and turnover are regulated, allowing renewal of the channel at the surface. In this review, we will deal primarily with the first two points, focusing on a Na_V_ channel-associated β subunit, namely β2. The β subunit family consists of four genes, *SCN1B*-*SCN4B*, which encode four different proteins, β1–4, of which β1 has two alternative splice variants, β1A and β1B; β subunits regulate sodium current (I_Na_) density [28]. Interacting with Na_V_1.5 by its extracellular region [29], or even through the transmembrane domain (TMD) [30], it has been proposed that they perform a major role mainly in ensuring efficient transport of the channel’s α subunit to the plasma membrane [13]. Interestingly, several BrS-associated mutations causing loss-of-function of the Na_V_ channel have been found in β subunits [31,32,33,34].

The current view on the function of β subunits within the Na_V_ channel is that (i) they determine proper cell surface localization of the channel complex, (ii) they control channel gating and kinetics, and also (iii) they regulate gene expression of the α subunit, and perhaps of the β subunits themselves [12]. The mechanisms by which β subunits may carry into effect these roles have been addressed in excellent recent reviews [35,36,37]. However, the data lead to the conclusion that modulation of Na_V_ channels by β subunits in heterologous systems shows effects that are dependent on the β and α subunits analyzed and, importantly, also of the cell type, which may be related to endogenous expression of some of the subunits. Indeed, it has been seen that in neurons or cardiomyocytes obtained from null mice, which would be more physiologically relevant models, the reported effects are in general less striking. Moreover, the protein domain(s) of β interacting with α, and posttranslational modifications, such as glycosylation and phosphorylation, clearly have an effect. In this regard, glycosylation of β subunits, and in particular sialylation, was found important for regulating the channel biophysical properties. Specifically, only sialylated β1 shifted gating of Na_V_1.2 and Na_V_1.5. On the other hand, sialylation of β2 influenced Na_V_1.5 gating, while not that of Na_V_1.2; those data also implicated that effects by these subunits on α are synergistic and, for β2 in particular, isoform specific [38].

The possibility that β subunits may compensate for each other in vivo adds another level of complexity to this aspect. In addition, Na_V_ β subunits can also influence certain potassium currents, thereby contributing to cross-talk between Na_V_ and potassium channels. Progress on this topic may be accomplished by investigating the structural determinants of the interaction and affinity between α and β subunits within the Na_V_ channel [37,39]. For instance, non-covalently bound β1 and β3 subunits affect the conformational dynamics of Na_V_1.5 by binding, through their extracellular and TMDs, to the voltage-sensing domains within domains III and IV of Na_V_1.5, thereby modulating ionic current kinetics and cell excitability [30].

Next, we will focus specifically on regulation of β2 trafficking, and how that could influence localization of Na_V_1.5, an aspect of key importance in cellular excitability and electrical coupling of the heart.

## 2. The β2 Subunit

### 2.1. Sequence and Domain Architecture of β2

Proper traffic and localization of β2 to subdomains of the plasma membrane is likely determined by its sequence motifs. Moreover, the interaction between β2 and Na_V_1.5, as well as with other channel-regulatory molecules, must be relevant to determine the correct localization of Na_V_1.5, and therefore the functionality of the Na_V_ channel. To examine this aspect, a biochemical dissection of β2 should be performed. β2 is a type I transmembrane protein with an extracellular, immunoglobulin (Ig)-like loop, having a role in cell adhesion [12], a single TMD, and a short cytoplasmic tail [40] (Figure 4). The extracellular loop, maintained by an intramolecular disulfide bond between Cys-50 and Cys-127 [29], has three potential *N-g*lycosylation sites, that is, Asn-42, Asn-66 and Asn-74 [38]. Within this region, a third cysteine, Cys-55, establishes a disulfide bond with the α subunit [29]. On the other hand, the short intracellular C-terminal domain has two possible phosphorylation sites, i.e., Ser-192 and Thr-204 [41] (based on UniProtKB accession number O60939, corresponding to human *SCN2B*).

### 2.2. Trafficking of β2 and Its Role within the Na_V_ Channel

The case of β2 is of particular interest concerning control of channel localization, since it is believed to influence localization of Na_V_1.5 at a post-Golgi site, on its targeting to the cell surface [42,43]. In addition, we have described the first mutation associated with BrS in *SCN2B* [32], the gene encoding β2. Our data have shown that the β2 D211G mutation (a substitution of Asp by Gly) causes an I_Na_ reduction of ~ 40% due to decreased cell surface levels of Na_V_1.5 [32,44]. Consistent with these data, it has been found that *Scn2b* deletion in mice entails, both in ventricular myocytes [45] and in primary cultures of hippocampal neurons [46], a comparable 40% decrease in surface levels of the α subunit and, as a result, of the I_Na_. Notably, β2 must associate with the α subunit for targeting to the nodes of Ranvier and the axon initial segment of the α/β2 complex [29].

We have shown that β2, exogenously expressed in Madin-Darby canine kidney (MDCK) cells, localizes in a polarized way at the apical plasma membrane domain [44]. In both MDCK cells and in cardiomyocyte-derived HL-1 cells, localization of Na_V_1.5 at the surface is strengthened by β2, but not by the subunit carrying the D211G BrS mutation. Regarding differential surface distribution, it is noteworthy the case of the β1 subunit, whose Tyr phosphorylation regulates its localization in sarcolemmal subdomains. As shown in mouse ventricular myocytes, Na_V_ channels at the ID consist of Na_V_1.5 and pYβ1, in close association with N-cadherin and Cx43, while channels in T tubules would be formed by other Na_V_ isoforms, such as Na_V_1.1 and Na_V_1.6, linked with unphosphorylated β1, and associated with ankyrin-B [26]. Understanding the mechanisms that determine these interactions is of the utmost importance, since the differential location of Na_V_ channels in subregions of the sarcolemma certainly influences conduction velocity and cardiac impulse propagation [20] (see Figure 3).

It is not known how β2 is transported to the cell surface; more specifically, it remains to be addressed how β2 is targeted preferentially to the apical region in MDCK cells. In fact, understanding how apical targeting signals are recognized in proteins is the focus of intense study. Such recognition can take place by association of the protein’s TMD with lipid rafts. It can also occur via *N*- or *O*-glycosylation of the luminal domain, and its consequent interaction with galectins (proteins that bind glycoprotein carbohydrates). Additional elements have been implicated, including certain Rab GTPases (belonging to the Ras superfamily of small GTPases), microtubule motor proteins, and elements of the actin cytoskeleton [47,48].

Based on these considerations, it is tempting to speculate that apical β2 localization may have a parallelism with its location at the nodes of Ranvier and the axon initial segment, where it interacts with the α subunit [29]. If β2 similarly distributes in specialized sarcolemmal regions, it would help to define, at least in part, the precise Na_V_1.5 location also at the cardiomyocyte surface. It is important to note that it remains to be investigated whether targeting of β2 to such surface regions, and its surface dynamics, potentially involving its endocytosis and turnover, are regulated by remodeling of the lipid bilayer, glycosylation, and/or dimerization/oligomerization. In this regard, we have recently found that *N-g*lycosylation of β2 is indeed necessary for its efficient trafficking and localization to the plasma membrane; importantly, non-glycosylated β2 does not promote surface localization of Na_V_1.5 [49].

### 2.3. Functions of β2

As demonstrated in *Scn2b* knockout mice, β2 has essential in vivo functions in maintenance of neuronal excitability. However, it was not found necessary for Na_V_ channel expression, or for survival, since knockouts develop normally, their neuronal function and morphology appear normal, and also their life expectancy. Yet, β2 is necessary for voltage-dependent inactivation of the I_Na_ and for maintenance of Na_V_ channel levels at the plasma membrane of the neuronal soma and in myelinated axons. The lack of β2 leads to hyperexcitability and to a decrease of the threshold to suffer seizures, possibly due to reduced excitability of inhibitory interneurons [46].

Concerning cardiac function, *Scn2b* knockout mice suffer from ventricular and atrial arrhythmias [45], which is consistent with *SCN2B* mutations described in humans [32,50]. In fact, these mice seem to reproduce some aspects of BrS, and it has been proposed that they could be very useful for modeling aspects of human ventricular arrhythmias with a greater susceptibility to atrial arrhythmia [45]. Intriguingly, a more recent report conclusively demonstrates that both β2 and β4 (unlike β1 and β3) are virtually undetected in ventricle of canine heart, implying that these subunits unlikely have any contribution to the regional manifestation of BrS, typically affecting the right ventricle rather than the left one [51].

Interestingly, human β2 can be sequentially cleaved by secretases; cleavage of β2 takes place analogously to the processing of the amyloid precursor protein (APP), whose amyloidogenic cleavage is cause and aggravation of the pathogenesis in Alzheimer’s disease [52]. In fact, β2 cleavage by α-secretase, and subsequently by γ-secretase, appears required for cell-cell adhesion and migration of β2-expressing cells [53]. In this regard, its extracellular domain, which confers β2 features of cell adhesion molecule (CAM), has been shown to interact via homophilic, and also heterophilic connections, by which it would bind the extracellular matrix proteins tenascin-R and tenascin-C, and also the β1 subunit, thereby functioning as a CAM [54,55]. Because β subunits are cell surface glycoproteins expressed during key periods of neuronal development, those earlier studies led to the hypothesis that they likely have a dual role, i.e., regulating the ion channel and functioning as CAMs. Such role on cell adhesion appears independent of their action on the channel α subunit, and would be responsible of bringing out an important connection between the extracellular environment and the cytoskeleton, for instance, by means of AnkG recruitment, which could have important implications in neural development [56].

The γ-secretase cleavage site in β2 [57], and probably that for β-secretase as well, appears to be conserved [58,59]. At least in mouse, the four β subunits (β1-4) can actually be processed by β-secretase, i.e., the β-site APP cleaving enzyme (BACE1) [59]. The cleaved intracellular domain of β2, as shown in neurons, is translocated to the nucleus to participate in transcriptional regulation of the α subunit, and possibly of other genes [52]. These results have been linked with observations from the null phenotype in mice. Since the *Scn2b* knockout mice displays uncontrolled fibrosis, i.e., cell replacement with fibrous connective tissue, it has been suggested that the β2 intracellular domain may regulate genes inhibiting atrial fibrosis [45].

Such remarkable finding reported over 10 years ago led to the conclusion that the β2 intracellular domain acts as a transcriptional activator for Na_V_1.1, although the increased α subunit accumulated intracellularly, while its cell surface levels were reduced [52]. Subsequently, these authors reported that BACE1 null mice have increased Na_V_1.2 surface levels, explained as a potential compensatory mechanism for the reduced surface Na_V_1.1 level. However, no changes in mRNA levels for Na_V_1.2 were detected [60]. This issue remains controversial as the phenotype of BACE1 null mice, i.e., susceptibility to seizures, could not be explained by alteration in expression and localization of Na_V_1.2 or Na_V_1.6, since protein levels of these α subunits remained unaltered and their distribution could not be correlated with the phenotype of these mice [61]. Interestingly, further work has shown that BACE1 can also have a non-enzymatic role on α, implying an influence on the electrical behavior of excitable cells independent of its proteolytic activity. Specifically, BACE1 was found to act as an accessory subunit of Na_V_1.2, mimicking β2 in the α/β channel complex. This feature of BACE1, i.e., acting as an accessory β subunit of Na_V_, and also potassium channels, would influence channel function, having an impact on cellular excitability and brain network activity, and at the same time may help explaining previous data. Importantly, this aspect opens a new area of study of BACE1 in physiology and pathology (reviewed in [62]).

Altogether, the published data provide strong evidence supporting a role of β subunits as signaling molecules involved in cell adhesion, migration, and neurite outgrowth [12]. In this regard, β1 was conclusively shown to promote neurite outgrowth in vivo through a signaling process triggered by trans-homophilic cell adhesion and association with lipid rafts, which was suggested to be essential for postnatal development of the central nervous system [63]. Within the context of cardiac function, an important result in this regard is the recent finding showing that β1, by mediating cell-cell adhesion, contributes to spreading of the AP along ventricular cardiomyocytes [64]. This work shows that β1 facilitates coupling of adjacent cells for ion exchange, being located within clefts of the perinexus, i.e., the narrow stretch of membranes closely apposed and adjacent to the gap junctions. By super-resolution optical microscopy, the authors concluded that β1 is located within Na_V_1.5-enriched nanodomains of the ID, however, it was seen clearly away from the fascia adherens areas rich in N-cadherin [64].

Here, we propose that β2 may have similar, yet to be discovered functions in adhesion and signaling also in cardiomyocytes. Investigating these potential features could help to understand better the role of β2 in specialized subdomains of the sarcolemma, both within and outside the Na_V_ channel. Indeed, β2 also binds laminin, which is the most abundant CAM in the extracellular matrix of the peripheral nervous system. In this regard, it has been suggested that β2 overexpression in prostate cancer mediates the neoplastic invasion of the nerves, which is believed to be a common pathway for cancer metastasis [65]. Intriguingly, this feature of β2 may have a connection with previous observation showing that β2 overexpression causes an increase of the cell surface membrane in Xenopus oocytes, which was then explained by increasing fusion of intracellular vesicles with the plasma membrane [40].

## 3. Overview and Working Hypotheses

In the context of the heart, the current data support the idea that β2 has a role in regulating localization and function of the Na_V_ channel in cardiomyocytes. In MDCK cells, a model system that we have used to imitate the polarized cardiomyocytes [18], β2 is located at the apical surface [44]. This distribution may have a correspondence with its preferential targeting to the neural nodes of Ranvier and the axon initial segment, where it interacts with the α subunit [29]. Given the importance of the α/β2 link, it is fundamental to understand how alterations in β2 may affect its localization and that of Na_V_1.5 at the cell surface. In this regard, we have analyzed the influence of β2 on localization and function of the Na_V_ channel in polarized MDCK cells. Our data show that targeting of β2 to the cell surface is affected if *N-g*lycosylation is abolished, causing retention of β2 in the ER. A small fraction of the triple non-glycosylated β2 mutant, that is, N42Q/N66Q/N74Q, can still reach the cell surface by bypassing the Golgi complex, and is correctly targeted to the apical domain. However, it is important to underline that it does not promote surface localization of Na_V_1.5, since coexpression of this mutant causes a reduction in surface levels of Na_V_1.5 [49] (Figure 5). Therefore, it is clearly defective in regulating Na_V_1.5 surface localization, which is a well-accepted function of β2 in the heart [12]. Yet, we have seen that a single *N-g*lycosylation is sufficient for β2 to reach the cell surface and to ensure efficient Na_V_1.5 localization [49]. To perform this role, we suggest that *N-g*lycosylation in β2 is necessary for its recognition through galectins, which would guide β2 to the proper cell surface domain [66].

For the β1 subunit, it has been proposed that palmitoylation is a possible mechanism for its adequate surface localization and consequent anchorage to lipid rafts [28]. Along these lines, we suggest that the TMD influences the localization of β2 within surface subdomains, which would also take place by association with these cholesterol-rich regions, and would possibly be regulated by the TMD length [67]. In this regard, our ongoing work shows that the apical distribution of β2 in MDCK cells is altered upon cholesterol depletion [68]. In primary cortical neurons, all four β subunits (β1-4) have in fact been found enriched in detergent resistant membranes by subcellular fractionation, implying their preferential association with lipid rafts [59]. On the other hand, it remains to be addressed the possible contribution of the Ig domain [29,69] in disulfide bond-mediated dimerization/oligomerization of β2 [70] and, consequently, in β2 targeting to certain surface domains.

We propose that mutations associated with β2—some of which undoubtedly linked to disease [32,50]—alter its dynamics at the membrane, negatively affecting localization and function of the Na_V_ channel. As previously seen [44,45,46], mutated β2, or its absence, negatively affects Na_V_1.5 traffic from the ER to the plasma membrane. A mutation in β2 may cause its intracellular retention. In the case of the BrS-associated D211G mutation, β2 reaches the plasma membrane normally. However, similar to unglycosylated β2 [49], this mutant is defective in promoting Na_V_1.5 trafficking to the surface [44]. In addition to β2 *N-g*lycosylation, which should ensure Na_V_1.5 chaperoning for proper folding, we propose that there are elements in the cytoplasmic domain of β2 potentially implicated on regulating Na_V_1.5 localization. In this regard, dynamics in β2 phosphorylation could potentially influence recruitment of adapter proteins, such as AnkG [19], thereby affecting targeting of the Na_V_ channel to the right surface subdomain. Given its proximity, the D211G mutation could influence potential phosphorylation predicted to take place in C-terminal residues [41] (see Figure 4). Support for this idea comes from previous work showing that the β2 intracellular domain, possibly interacting with AnkG, allows the heterophilic interaction with the extracellular domain of β1, thereby stabilizing extracellular cell adhesion. The region in β2 required was mapped, and encompasses two Thr residues, i.e., T198 and T204, that belong to a consensus sequence for potential phosphorylation by casein kinase II, and thereby may be critical for AnkG interaction, in analogy to Tyr-phosphorylated β1 [71,72].

Additional evidence suggesting that there is an indirect link between Na_V_1.5 and β subunits implicated on regulating localization of α also comes from earlier work showing that the cytoplasmic domain of β subunits may not have much influence on α/β interaction in heterologous cell systems. Thus, a β1 chimera bearing the intracellular domain of β2 overlapped strongly with Na_V_1.5, supposedly in intracellular compartments [73], similarly as β1 does, but in contrast to β2, which is efficiently targeted to the cell surface [42]. Moreover, recent data obtained by cryo-electron microscopy confirmed that direct α/β2 interaction takes place essentially through extracellular disulfide binding, i.e., at Cys-55, as previously shown [29], in addition to hydrogen bonding with Tyr-56 and Arg-135, although the latter, connecting with the Ig loop, are likely less stable interactions [74,75]. Interestingly, a previous report presenting 3D structural data of the β2 extracellular region obtained by X-ray crystallography revealed the exact points of contact with the α subunit and indicated that β2 is unique among the Na_V_ β subunits in that it contains a second intrasubunit disulfide bond, via Cys-72 and Cys-75, in addition to the common Cys-50 and Cys-127 bond, conserved in its β counterparts, yet this peculiarity may have a yet to be elucidated function [76]. On the whole, we suggest that, at least in part, the action of β2 on α, and in particular on Na_V_1.5, may take place indirectly.

## 4. Limitations: The Cell Model System

Carrying out studies in immortalized cell lines, such as Human Embryonic Kidney 293 cells, Chinese Hamster Ovary cells, or MDCK cells, represents an obvious limitation to recapitulate the biology of the β2 subunit in the cardiomyocyte, or in other excitable cells. Like cardiomyocytes, MDCK cells are polarized, but unlike those, they are epithelial cells. On the other hand, HL-1 cells are of cardiac origin, albeit they are not polarized. In general, mechanisms governing trafficking have been studied in cell lines growing in artificial environments, such as monolayers attached on a substrate, or embedded in a matrix resembling the extracellular milieu. These setups are suitable for analyzing many aspects addressed here. However, most of these cell lines have the obvious limitation that they are not excitable. Even HL-1 cells, which have endogenous I_Na_ [77], do not express β subunits and have very low levels of Na_V_1.5 ([44], and our unpublished data), and must also be transfected to study sorting and targeting of the Na_V_ channel. Likewise, most of these cell lines probably lack specific channel-associated proteins, as well as components of the machinery required for sorting, targeting, or membrane anchoring of channel subunits and, therefore, for their proper localization in the cardiomyocyte. Therefore, we must keep in mind that when studying the role of β subunits, their trafficking, or interaction with α, it is quite possible that some observed effects differ from what really occurs in a cardiomyocyte or in other excitable cells. Definitely, this issue should be addressed by researchers in the field, and a considerable effort must be put on developing more suitable models to analyze trafficking of the Na_V_ channel subunits and also of the channel’s biophysical properties.

## 5. Implications of Research on the Na_V_ Channel β Subunits

Research projects strictly focused on the biological aspects, as those with a strong relevance to clinical aspects, are valid to address the aspects discussed here. Studying the mechanisms that may be altered in heart disease poses to address one of the main challenges of society, that is, human health. A key hypothesis is that alterations in localization of components of the Na_V_ channel complex convey a risk of arrhythmia and, therefore, are potentially associated with important cardiac pathologies.

Altogether, it is of utmost importance that findings in the field lead to a better understanding of how subunits of the Na_V_ channel correctly localize. Undoubtedly, growing knowledge on this aspect will provide a better picture on the link between cell excitability and the electrical coupling in the heart, thereby contributing to a better knowledge of how arrhythmias develop.

In summary, future studies should be aimed to understanding better the genetic factors that determine phenotype variability. We are convinced that data generated in this field will be very useful for risk stratification of the patient and therefore to predict heart disease. This should contribute to early diagnosis of disease in asymptomatic individuals potentially at risk of SCD and to improve treatment therapies, also avoiding a fatal event in patients already suffering from heart disease.

## Figures and Tables

**Figure 1 biomolecules-09-00604-f001:**
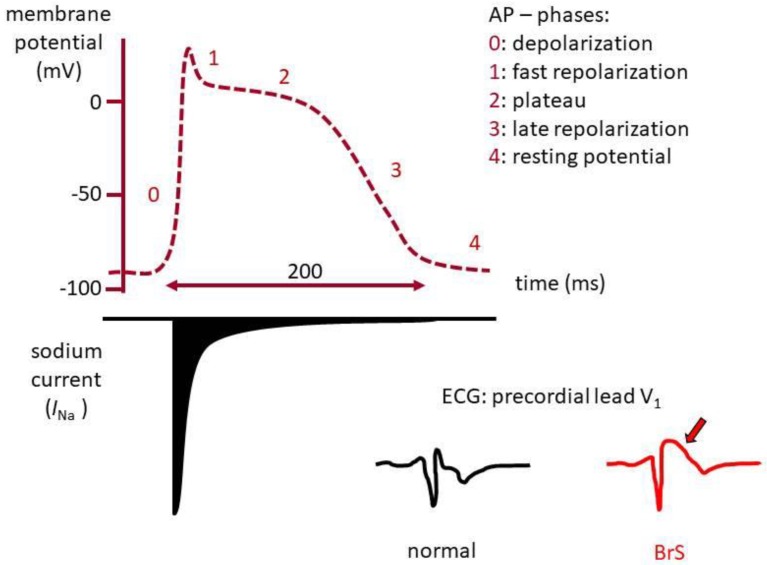
Role of Na_V_1.5 in the generation of the cardiac action potential (AP). Schematic representation of the cardiac AP and the contribution of the sodium current (I_Na_) generated by Na_V_1.5 as responsible for the rising (depolarization) phase. The electrical pattern of precordial lead V1 of a normal electrocardiogram (ECG) and one with an ST segment elevation typical of BrS are shown (arrow).

**Figure 2 biomolecules-09-00604-f002:**
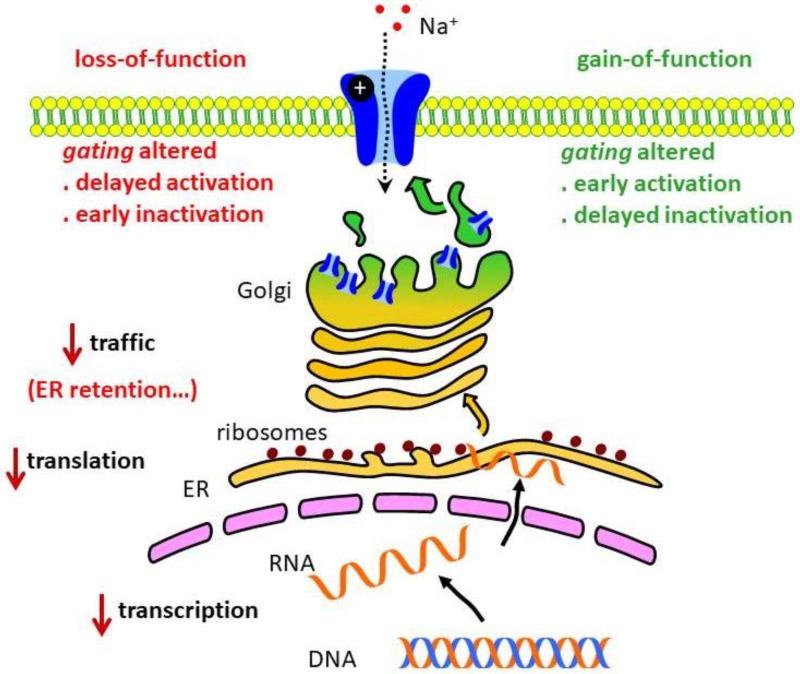
Alterations in the Na_V_ channel associated with heart disease. Several molecular mechanisms can affect expression, localization or function of the channel and are associated with inherited or acquired heart disease. ER, endoplasmic reticulum.

**Figure 3 biomolecules-09-00604-f003:**
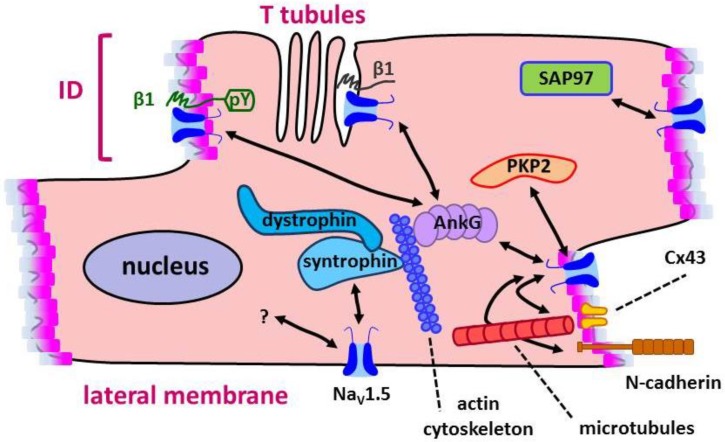
Multiple interactions implicating Na_V_1.5 in cardiomyocytes. Several proteins have been identified interacting with Na_V_1.5 in different subregions of the cardiomyocyte sarcolemma, establishing protein complexes that define different pools of Na_V_1.5. See the text for details.

**Figure 4 biomolecules-09-00604-f004:**
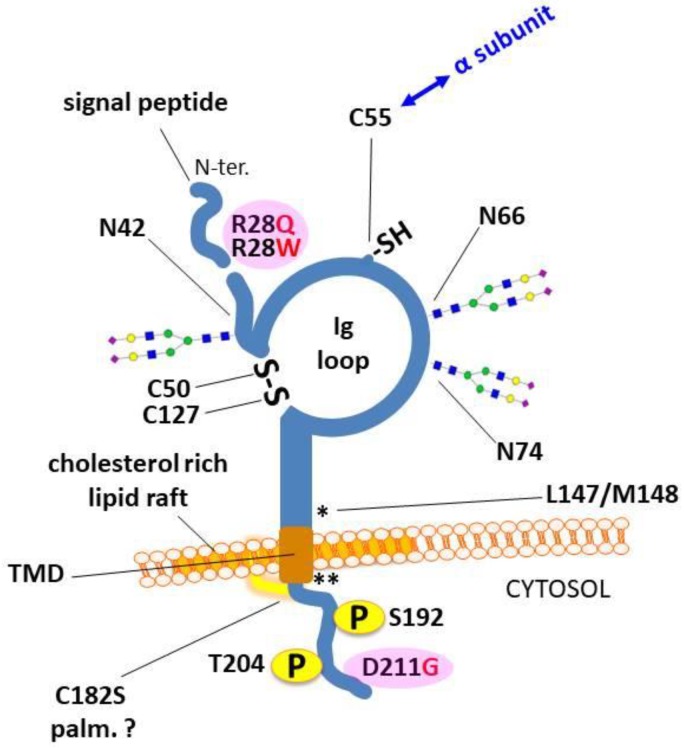
Sequence and domain architecture of the β2 subunit. Of its 215 amino acids, the first 29 correspond to the predicted N-terminal (N-ter.) signal peptide [41]. It is worth noting the extracellular immunoglobulin (Ig)-like loop. The transmembrane domain (TMD), which expands along ~ 20 residues, and probably associates with lipid rafts, is followed by a short C-terminal domain. Asterisks denote the approximate location of cleavage sites by β-secretase (*) and γ-secretase (**). Pathogenic mutations, i.e., the atrial fibrillation-associated R28Q and R28W, and the BrS-associated D211G, are shown in pink background; see the text for more details.

**Figure 5 biomolecules-09-00604-f005:**
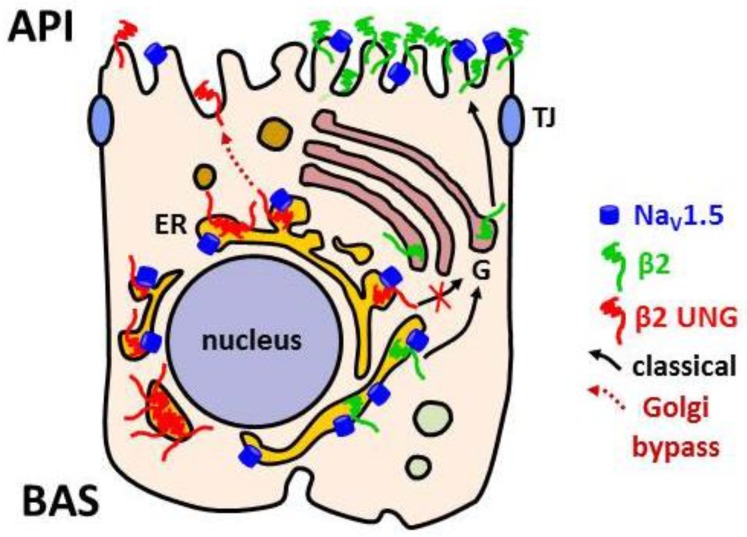
*N-g*lycosylation of β2 is necessary for its targeting to the cell surface and to promote surface localization of Na_V_1.5. Proper β2 trafficking is determinant for Na_V_1.5 localization to the cell surface, and specifically, to the apical plasma membrane (API) in polarized MDCK cells. Unglycosylated (UNG; N42Q/N66Q/N74Q) β2 is retained in the endoplasmic reticulum (ER), often seen together with Na_V_1.5. By bypassing the Golgi apparatus (G), a fraction of the mutant can reach the cell surface (dotted arrow), but at a rate of only approximately one-third of that of the wild type. In addition, it is defective in promoting surface localization of Na_V_1.5. BAS, basolateral surface; TJ, tight junctions.

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
