# Peer review of "Trafficking and Function of the Voltage-Gated Sodium Channel β2 Subunit"

_biomolecules, 2019, doi:10.3390/biom9100604_

Round 1
Reviewer 1 Report
well written review
Reviewer 2 Report
Role of subunits in cell adhesion was mentioned but did not clearly discussed in later parts Introduction is helpful but it is too general. Could try to focus on the mechanisms and regulation aspect of the beta subunit Figures are good and understandable Certain places like page 5 line 173 you said we have described.. but there is no reference Certain sentences are not clear for example page 6 line 201; page 7 line 235 You mentioned that cleaved intracellular domain of beta2 acts as a transcription factor. I believe this aspect is not much discussed in the field. Can you elaborate on this?Author Response
Please see the attachment
